# EFFICIENT MULTI-LEVEL LEARNING FOR DENSE OBJECT DETECTION

## ABSTRACT

Dense object detection is crucial and favorable in the industry and has been popular for years with the success of the multi-level learning framework. By delivering the learning of objects into a multi-level feature pyramid, such a divide-and-conquer solution eases the optimization difficulty. However, this learning paradigm has a major shortcoming left behind. The shallow levels take tons of computational burden due to their high resolutions of the feature maps, heavily slowing down the inference speed. In this paper, we aim for minimal modifications to exchange a better speed-accuracy trade-off. The outcome is SlimHead, a very simple, efficient, and generalizable head network, which further unleashes the potential of multi-level learning for dense object detectors. It operates in two stages: Slim and Fat, initially plugging interpolator before the head network functions to "slim" the feature pyramid, and then recovering the features to original solution space by "fatting" the feature pyramid. Thanks to its flexibility, operations with higher computational complexity can be easily integrated to benefit accuracy without loss of inference efficiency. We also extend our SlimHead to multiple high-level vision tasks such as arbitrary-oriented object detection, pedestrian detection, and instance segmentation. Extensive experiments on PASCAL VOC, MS COCO, DOTA, and CrowdHuman demonstrate the broad applicability and the high practical value of our method. All the source code and tutorials will be made publicly available.

## 1 INTRODUCTION

Dense object detection is a long-standing research topic in computer vision and continues to have a positive impact on relevant fields, such as arbitrary-oriented object detection (Yang et al., 2019; Yu & Da, 2023), pedestrian detection (Shao et al., 2018), and instance segmentation (Tian et al., 2021; 2020). Until now, it still holds an unshakable dominant position in industrial applications due to its excellent speed-accuracy trade-off and friendliness to low-end edge devices. Objects vary in size. The community therefore comes to a solution of multi-level learning that delivers the learning of large objects to deep levels (feature maps with low resolution) and the learning of small ones to shallow levels (feature maps with high resolution). A typical example is FPN-like methods (Lin et al., 2017a), building a multi-level feature pyramid to process the backbone features and predict the instances in parallel at multiple levels. This learning paradigm has been validated to be effective and thus becomes dominant in the field of dense object detection (Lin et al., 2017b; Kong et al., 2020; Tian et al., 2019; Zhang et al., 2020; Li et al., 2020; Feng et al., 2021).

Recently, a bunch of research breakthroughs in dense object detection focus on enhancing the consistency between classification and localization (Jiang et al., 2018; Li et al., 2020; Zhang et al., 2021; Li et al., 2022b; Feng et al., 2021), alleviating localization ambiguity (He et al., 2019; Choi et al., 2019; Li et al., 2020; Zheng et al., 2022; 2023b), as well as improving localization quality (Rezatofighi et al., 2019; Zheng et al., 2020; 2021; Wang et al., 2020). Although multi-level learning is the foundation of all the above works, there is still one major shortcoming left behind, which is common sense yet little attention has been paid to: *The shallow level in high resolution is too time-consuming.* A problem caused by the defect is that the head networks occupy a large proportion of computations, even though they are lightweight in terms of model parameters. As shown in figure 1, a head network with only 15.6% parameters can produce 52.0% FLOPs (floating point operations). Furthermore, the multi-levels are parallel, which indicates that the operations are usually shared

between different levels. This makes heavy operators (e.g., deformable conv (Zhu et al., 2019)) that are beneficial for accuracy improvement more computationally burdensome.

In this paper, we rethink the multi-level learning paradigm by investigating performance sensitivity, where we delve into the intrinsic property of the head networks and find out which components are essential to accuracy and speed. A natural outcome of our exploration is a very simple, efficient, and generalizable head network, termed SlimHead, for dense object detection. In our design ethos, SlimHead operates in two stages: Slim and Fat. In the first stage Slim, we inject an interpolator before the head network functions to "slim" the feature pyramid. This produces a compact and efficient head network. Then, in the second stage Fat, we employ the inversed interpolator to "fat" the feature pyramid. As such, the features are recovered to the original solution space. We find that this key difference between our SlimHead and the traditional head networks is essential to reducing computations and holding accuracy. We show that when integrated correctly, this plug-and-play strategy elegantly aligns the solution space of predictions and therefore none of extra modifications are required. As a result, SlimHead enables us to notably alleviate the computational burden of the head networks while maintaining comparable accuracy. Furthermore, operations with higher computational complexity (e.g., deformable conv (Zhu et al., 2019)) can be effortlessly integrated to achieve accuracy gains without loss of efficiency. As a fringe benefit, our method can also save GPU memory usage (reduced by 15.1% on ResNet-18), which is more user-friendly for deployment on low-end edge devices.

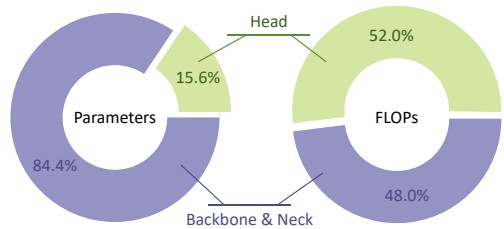

Figure 1: The head networks occupy a large proportion of computational complexity, even though they are lightweight in terms of model parameters. This problem is common among dense object detectors.

The highlights of this paper are two-fold:

- We reawaken the issue of the efficiency bottleneck in dense object detection. The outcome of our exploration is a very simple, efficient, and generalizable head network, termed SlimHead, which achieves a better speed-accuracy trade-off. Thanks to its flexibility, operations with higher computational complexity can be effortlessly integrated to achieve accuracy gains without loss of efficiency. The superiorities of our SlimHead: *Better accuracy, faster speed, easy-to-implement, and lower GPU memory usage.*

- We extend our SlimHead to multiple high-level vision tasks, e.g., arbitrary-oriented object detection, pedestrian detection, and instance segmentation. The results on PASCAL VOC (Everingham et al., 2010), MS COCO (Lin et al., 2014), DOTA (Xia et al., 2018), and CrowdHuman (Shao et al., 2018) demonstrate the broad applicability and the high practical value of our method.

## 2 BACKGROUND

### 2.1 MULTI-LEVEL LEARNING IN OBJECT DETECTION

Multi-level learning, also referred to as the neck+head network, is a conventional paradigm to detect objects with various sizes in a manner of feature pyramids (Ren et al., 2015; Liu et al., 2016; Redmon & Farhadi, 2018; Lin et al., 2017b; Bochkovskiy et al., 2020; Jocher et al., 2022). Prosperous research progress has been made on how to build stronger neck networks. For example, the pyramidal feature hierarchy follows the principle of the bottom-up pathway. Typical approaches such as SSD (Liu et al., 2016) and STDN (Zhou et al., 2018) directly leverage multiple feature levels as the prediction layers. Since FPN (Lin et al., 2017a), the feature fusion has gotten more and more research interest. The basic idea is to deliver the knowledge of deep feature levels to the shallow feature levels again by top-down pathway and lateral connections. DSSD (Fu et al., 2017), PANet (Liu et al., 2018), NAS-FPN (Ghiasi et al., 2019), and Bi-FPN (Tan et al., 2020), respectively studied the deconvolution, bottom-up path augmentation, Neural Architecture Search (NAS), and repeated bi-directional feature pyramids for building a powerful neck network. ASFF (Liu et al., 2019) proposed to conduct weighted spatial feature fusion for each level. In (Kong et al., 2018), nonlinear global attention is proposed to reconfigure the deep feature pyramid. QueryDet (Yang et al., 2022a) added a P2 level, query head,

and sparse conv for the speed-accuracy trade-off, but it needed some necessary hand-crafted designs for using sparse conv ops and must search for a better hyper-parameter for the loss functions again as the prediction maps change. YOLOF (Chen et al., 2021a) built a single-level dense object detector. While YOLOF successfully reduces the computational burden, it relies on some tailored designs for single-level models, e.g., stacked dilation blocks and uniform matching. This makes it difficult to generalize single-level models to popular multi-level ones.

## 2.2 HEAD NETWORKS IN OBJECT DETECTION

Head networks are commonly used for further refining the features from the upstream networks. The typical components are stacked convolution ops. The number of conv ops usually ranges from 1 (RPN (Ren et al., 2015)) to 6 (TOOD (Feng et al., 2021)), among which, 4 conv layers are widely adopted, e.g., RetinaNet (Lin et al., 2017b), FCOS (Tian et al., 2019), etc. Some object detectors adopt the full-connect (FC) layer in head networks, e.g. R-CNN series (Ren et al., 2015; Cai & Vasconcelos, 2018; Pang et al., 2019; He et al., 2017). Another special is Double-Head (Wu et al., 2020), which empirically observes the FC layer is suitable for the classification branch while the localization branch favors the conv layer more. Dynamic Head (Dai et al., 2021a) considered 3 kinds of attention in the head network, i.e., scale-, spatial-, and task-aware. GFocal (Li et al., 2020) proposed to jointly optimize classification and localization and removed the center-ness branch proposed by FCOS. DDQ-FCN (Zhang et al., 2023) integrated channel fusion in the head while GFocalV2 (Li et al., 2021) added an FC module to predict the localization quality estimation. PAA (Kim & Lee, 2020) leveraged score-voting NMS (Non-Maximum Suppression) while VFNet (Zhang et al., 2021) proposed box refinement with star-shape box feature representation. Some approaches proposed better bounding box representation for capturing the localization ambiguity, e.g., Gaussian distribution representation (He et al., 2019; Choi et al., 2019) and probability distribution representation (Li et al., 2020; Qiu et al., 2020), thereby enhancing the localization quality. There are also some methods to improve the detection performance without losing inference efficiency, e.g., label assignment (FreeAnchor (Zhang et al., 2019), ATSS (Zhang et al., 2020), PAA (Kim & Lee, 2020), OTA (Ge et al., 2021), DW (Li et al., 2022b), SELA (Zheng et al., 2023a)), loss function (GHM (Li et al., 2019), and IoU-based losses (Yu et al., 2016; Rezatofighi et al., 2019; Zheng et al., 2020; 2021; He et al., 2021)), and knowledge distillation (LD (Zheng et al., 2022; 2023b), FGD (Yang et al., 2022b), PKD (Cao et al., 2022), CrossKD (Wang et al., 2024)).

In the above methods, the operations in the head networks are usually parallel across feature levels. Little attention is paid to the imbalanced efficiency between different feature levels. In this work, we rethink multi-level learning and propose a new SlimHead method to balance the computational complexity between the shallow levels and the deep ones.

## 3 METHODOLOGY

Our goal is to search which components are essential in building an efficient and powerful head network. Through a series of experiments, the outcome is SlimHead, a very simple, efficient, flexible, and generalizable head network for dense object detection.

### 3.1 ANALYSIS ON MULTI-LEVEL LEARNING

**Overall structure analysis.** Multi-level learning is defined as a parallel optimization problem:

$$\min_{\Theta} \sum_i \mathcal{L}(\mathcal{H}(X_i|\Theta), G_i), \qquad (1)$$

where $i$ is the level index, $X_i$ is the features at level $i$, $\mathcal{H}$ is the head network functions with shared parameters $\Theta$, $G_i$ is the corresponding ground-truth supervision at level $i$, and $\mathcal{L}$ is the given loss function. A typical form of multi-level learning in dense object detection can be seen in figure 2, which consists of 5 feature levels from P3 to P7 and has been widely adopted in the popular dense object detectors, e.g., RetinaNet (Lin et al., 2017b), FCOS(Tian et al., 2019), GFocal (Li

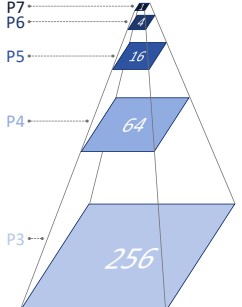

Figure 2: Feature pyramid with actual size. Shallow levels occupy tons of computation. The computation of each level is 4 times that of the deeper one.

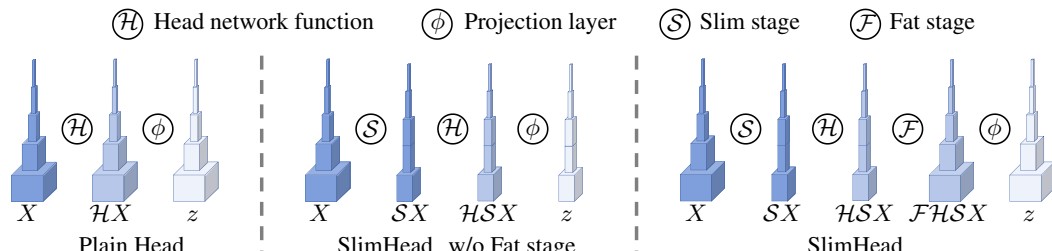

Figure 3: A typical multi-level learning of head network (**Left**) *vs.* our SlimHead (**Right**). $X$: Features from the upstream network, e.g., backbone & FPN. $z$: Output logits for both class scores and bounding boxes. Plain Head (**Left**) consists of 5 pyramid levels from $P_3$ to $P_7$. SlimHead w/o Fat stage (**Middle**): We only introduce the Slim stage to slim the features $X$, which releases the computational burden of shallow levels. SlimHead (**Right**): we further employ the Fat stage which transforms the feature $\mathcal{HSX}$ back to its original dimention.

et al., 2020), TOOD (Feng et al., 2021), etc. It is easy to see that for such an optimization problem, three terms will affect the model capability.

1. The network function $\mathcal{H}$ - maps the semantic features to logits with specific physical meanings. Generally, $\mathcal{H}$ consists of a series of stacked convolutions to refine the features. It is noteworthy that $\mathcal{H}$ is usually parallel across different levels since the parameters are shared.

2. The input features $X_i$ - affect the whole optimization process and model efficiency.

3. The loss function $\mathcal{L}$ - determines the optimization direction. This is also parallel between levels.

Let's delve deeper into features $X_i$. Firstly, $X_i \in \mathbb{R}^{b \times C \times H_i \times W_i}$ is the output features from the upstream network, i.e., backbone & FPN, where $b$ and $C$ are the batch size and the number of channels, $W_i$ and $H_i$ are the width and the height of feature map. The resolution of $X_i$ determines the inference speed. The higher the resolution of feature maps, the slower the inference speed, and vice versa. Secondly, the resolution of $X_i$ also determines the number and the size of anchors, which has a significant impact on the accuracy. Since $\mathcal{L}$ is not about inference speed, in the following, we investigate the sensitivity of $\mathcal{H}$ and $X_i$ to accuracy and speed.

**Plain Head analysis.** The plain head is the foundation of the current dense object detectors (Lin et al., 2017b; Tian et al., 2019; Zhang et al., 2020; Li et al., 2020; 2021; Zhang et al., 2021; Li et al., 2022b; Feng et al., 2021), as shown in figure 3**Left**. Given a head network function $\mathcal{H}$ and a projection layer $\phi$, the multi-level features $X = \{X_i\}$, $i = 3, 4, 5, 6, 7$, from the upstream networks will be refined and projected to the output logits $z = \phi \mathcal{H} X$. A naive approach to reducing the computational burden of the head network is to reduce the number of conv layers. In figure 4, we reduce the conv layers one by one. It shows that the detection accuracy degrades as the number of conv layers decreases, though the inference speed becomes faster. This indicates that the head network needs multiple

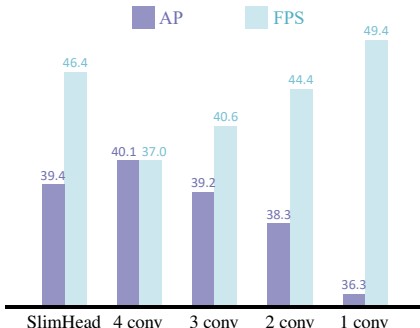

Figure 4: Accuracy and speed versus number of conv layers.

conv ops to refine the features. Using fewer conv ops in head networks cannot reach a good accuracy. We wonder which components in the head network are essential to achieving a better speed-accuracy trade-off.

## 3.2 SLIMHEAD

According to figure 2, the shallow levels are too time-consuming, limiting the efficiency of dense object detectors. In this paper, we encourage exploring new insights on the intrinsic property of head networks. We observe that the feature dimention can be shrunk for getting inference speedup while maintaining comparable accuracy as long as the logit map dimention is hold. To keep the context flow, we introduce the proposed SlimHead first. There are two stages in our design ethos. In the following, we will delve into the Slim and Fat stages.

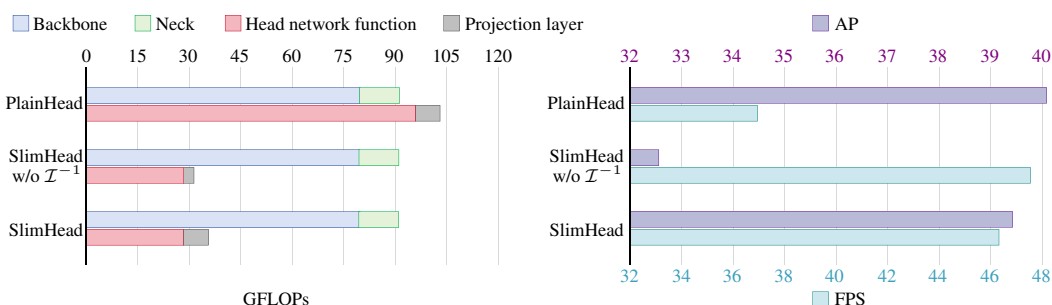

Figure 5: GFLOPs, accuracy (AP), and speed (FPS) comparison between various head network designs. Our SlimHead significantly reduces the computational complexity for the head networks. Meanwhile, we achieve a comparable accuracy of 39.4 AP and 46.4 FPS (25.4% speedup). In the middle group, it shows a severe AP drop if we do not transform the features back to their original sizes.

**Stage-I: Slim**. We propose to inject an interpolator $\mathcal{S}$ with a scaling ratio $r$ into the head network before refining features by the head network functions $\mathcal{H}$. Note that this strategy can be conceptually applied to any level, but we found it unnecessary for deeper levels since they are not the efficiency bottleneck of the model. Thus, we introduce a level selector $K \in \{3, 4, 5, 6, 7\}$ to select $i \leqslant K$ levels to apply the Slim stage:

$$\mathcal{S}X = \mathcal{S}(X_i; r), \quad i \leqslant K. \tag{2}$$

**Stage-II: Fat**. In the Fat stage, we further inject an inversed interpolator $\mathcal{F}$ with a scaling ratio $1/r$ before the projection layer, which can be written as

$$\mathcal{F}\mathcal{H}\mathcal{S}X = \mathcal{F}\left(\mathcal{H}(\mathcal{S}(X_i; r)); \frac{1}{r}\right), \quad i \leqslant K. \tag{3}$$

This enables us to transform the features back to their original dimention, which guarantees the same number of predicted boxes for each anchor location. All the hyperparameters involved in the training process keep consistent, e.g., the same anchor definition, the same label assignment, and the same hyperparameters in loss functions. As a result, the solution space of optimization remains unchanged. The illustration of SlimHead can be seen in figure 3**Right**.

There are four appealing advantages of the proposed SlimHead: 1) The Slim stage makes the feature pyramid $X$ slimmer, which substantially release the computational burden of shallow levels. When the scaling ratio $r = 1$, SlimHead degrades to the original head networks. For cases where $r < 1$, the computational complexity of the level is reduced to $r^2$ of the original. 2) As the computational complexity is reduced, operations with higher computational complexity can be integrated without severe speed degradation, e.g., deformable conv (Zhu et al., 2019). 3) As the feature dimention of the shallow levels decreases, the GPU memory usage can also be notably reduced. 4) SlimHead is highly generalizable. In most previous methods (Lin et al., 2017b; Li et al., 2020; Tian et al., 2019; Feng et al., 2021), no matter what feature aggregation operations are used, our method can be easily incorporated into the dense object detectors.

**SlimHead analysis.** To investigate the intrinsic property of the head networks, we compare two SlimHead variants. The first one adopts the Slim stage only, which we call "SlimHead w/o Fat stage", as shown in figure 3**Middle**. The second is the full version of SlimHead (figure 3**Right**). In figure 5, we showcase the performance sensitivity of the proposed SlimHead. It can be seen that both SlimHeads largely reduce the computational complexity of object detector. Intriguingly, "SlimHead w/o Fat stage" produces severe accuracy drops of about $-7.5$ AP. If we add the Fat stage, our full version of SlimHead can achieve a comparable accuracy of 39.4 AP *vs.* 40.1 AP (baseline) while gaining a speedup of 25.4 %. This indicates that it is necessary to keep the solution space of optimization since all the hyper-parameters involved in label assignment and loss function are tailored based on the sizes of the output logit maps.

In figure 4, our SlimHead shows promising results that can achieve considerable speedup while maintaining high detection accuracy. More attempts to reduce the computational burden of multi-level learning can be found in the Appendix (section A.2), despite we found that our SlimHead is the most simple and effective among them.

Table 1: Ablation with different interpolation functions. We study 3 interpolation modes: "nearest", "max-pool", and "bilinear".

| Mode | AP | $AP_{50}$ | $AP_{75}$ | FPS |
|------|-----|-----------|-----------|------|
| baseline | 40.1 | 58.2 | 43.1 | 37.0 |
| nearest | **39.7** | 57.8 | **42.8** | **43.9** |
| max-pool | 39.7 | **58.0** | 42.7 | 43.2 |
| bilinear | 39.5 | 57.8 | 42.5 | 43.7 |

Table 2: Ablation with different level selector $K$.

| $K$ | AP | $AP_{50}$ | $AP_{75}$ | FPS |
|------|-----|-----------|-----------|------|
| baseline | 40.1 | 58.2 | 43.1 | 37.0 |
| 3 | **41.4** | **59.4** | **44.8** | 38.0 |
| 4 | 41.1 | 59.2 | 44.3 | 40.7 |
| 5 | 40.6 | 58.9 | 43.7 | 41.3 |
| 6 | 40.0 | 58.5 | 43.4 | 41.3 |
| 7 | 39.6 | 58.2 | 42.9 | **41.9** |

Table 3: Ablation with different scaling ratio $r$. DCN denotes that we apply DCN (Zhu et al., 2019) to the first two layers of the head networks.

| $r$ | DCN | AP | $AP_{50}$ | $AP_{75}$ | FPS |
|------|------|-----|-----------|-----------|------|
| 1.0 | | **40.1** | **58.2** | **43.1** | 37.0 |
| 0.9 | | 39.2 | 57.7 | 42.4 | 38.0 |
| 0.8 | | 39.2 | 57.6 | 42.1 | 39.9 |
| 0.7 | | 39.6 | **58.0** | 42.6 | 40.6 |
| 0.6 | | 39.2 | 57.6 | 42.4 | 42.2 |
| 0.5 | | **39.7** | 57.8 | **42.8** | 43.9 |
| 0.4 | | 37.9 | 56.3 | 40.2 | **45.1** |
| 1.0 | ✓ | **42.0** | **60.0** | **45.6** | 29.9 |
| 0.9 | ✓ | 40.9 | 59.3 | 44.3 | 31.7 |
| 0.8 | ✓ | 40.9 | 58.9 | 44.2 | 34.0 |
| 0.7 | ✓ | 41.0 | 59.2 | 44.4 | 35.2 |
| 0.6 | ✓ | 40.7 | 58.9 | 44.1 | 36.8 |
| 0.5 | ✓ | **41.4** | **59.4** | **44.8** | 38.0 |
| 0.4 | ✓ | 39.6 | 57.9 | 42.3 | **39.6** |

# 4 EXPERIMENTS

In this section, we conduct experiments on the challenging MS COCO benchmark (Lin et al., 2014). The train / val sets are COCO train2017 (118K images) and val2017 (5K images), respectively. We report the COCO-style average precision (AP) as the main metric. Since our method is proposed for efficient object detection, we also report Frame Per Second (FPS) for evaluating the inference speed. The inference speeds are measured on a single RTX 3090 GPU for all detectors. We adopt the MMDetection (Chen et al., 2019) framework. All hyper-parameters except the scaling ratio $r$ and the level selector $K$ remain unchanged for a fair comparison. In the ablation study, we adopt the popular one-stage object detector GFocal (Li et al., 2020) with ResNet-50 (He et al., 2016) backbone and FPN (Lin et al., 2017a) neck as the baseline. Unless otherwise stated, the classic single-scale $1\times$ (12 epochs) training schedule with $1333 \times 800$ resolution is adopted by default. $2\times$ training schedule indicates we train the network for 24 epochs with multi-scale training $[480 : 960]$.

## 4.1 ABLATION STUDY

**The interpolation function $\mathcal{I}$.** We study 3 interpolation functions $\mathcal{I}$. The first one is nearest neighbors interpolation. The second is the max-pooling algorithm. The third is bilinear interpolation. In this experiment, we only apply $\mathcal{I}$ and $\mathcal{I}^{-1}$ at the shallowest level, i.e., the level selector $K = 3$. The results are reported in Tab. 1. It can be seen that our method can achieve detection accuracy comparable to the baseline model, while also being more efficient. Among the 3 interpolations, the nearest one is simple and efficient, which also reaches the highest accuracy. Thus, in the following experiments, we adopt the nearest neighbors interpolation by default.

**The level selector $K$.** The core of SlimHead can be conceptually applied to any level. We conduct an experiment to observe the changes in detection performance when applying SlimHead to pyramid levels from shallow to deep. Benefit by the efficiency of SlimHead, we adopt DCN (Zhu et al., 2019) at the first two layers. Tab. 2 shows that our method can achieve a better and faster performance when $K \leqslant 5$, i.e., the shallow levels. We find that the detection performance degrades when we apply SlimHead to deeper levels. Also, it does not benefit inference speed much, as deep levels are not a bottleneck in computational complexity. Therefore, in practice, we usually apply SlimHead at shallow levels.

**The scaling ratio $r$.** When applying SlimHead, the shallow-level features will be temporarily transformed into a small feature space to reduce computational complexity. We study the impact of $r$ and the results are reported in Tab. 3. In this experiment, we set $K = 3$, i.e., we only apply our SlimHead to the shallowest level. One can see from the first group in Tab. 3 that our method can achieve accuracy comparable to the baseline model ($r = 1$). Particularly, when $r = 0.5$, we achieved an FPS of 43.9, accelerating the inference speed of the detector by nearly 20.3%. To go one step further, we incorporate DCN (Zhu et al., 2019) into our SlimHead, where only the first two layers are

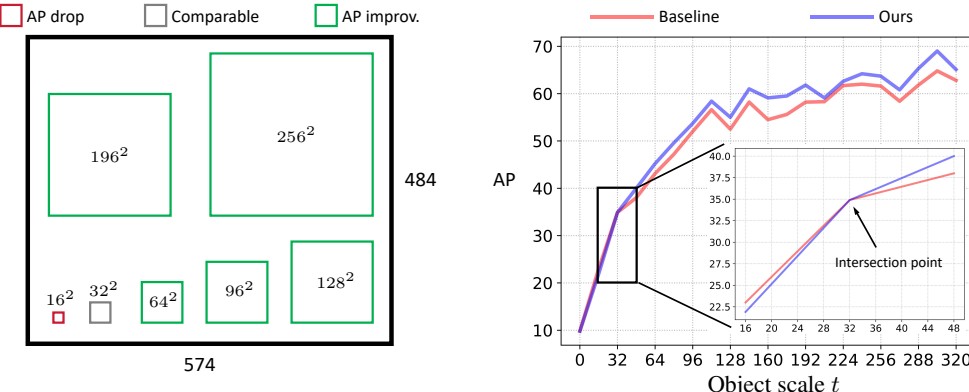

Figure 6: **Left:** The effect of SlimHead on objects of different scales. The average size of the image is shown by a black box. **Right:** The AP curves of objects whose box areas fall in the interval $[t^2, (t+16)^2]$.

replaced. The second group in Tab. 3 shows us a promising improvement in detection accuracy. If we directly replace the convolution with DCN on the original head networks ($r = 1$), the detector will significantly slow down to a speed of 29.9 FPS. Noticeably, our SlimHead achieves 41.4 AP and 38.0 FPS at $r = 0.5$, which is even better and faster than the baseline model (the 1-st row). It shows that we can improve the object detector by + 1.3 AP while gaining speed acceleration for free, which was unaffordable in previous methods because shallow levels take up a large proportion of the computational burden, especially for the improved conv ops, e.g., DCN. Besides, it is worth noting that our method achieves a peak of AP improvement at $r = 0.5$. This suggests that it would be better to transform the shallow level to a size similar to the neighboring one. In the following, we set $r = 0.5$ by default.

**The effect of SlimHead on objects of different scales.** Since our method changes the feature map size of the head networks, it may produce different effects on objects of different scales. Firstly, we report the $\mathbf{AP}_S$, $\mathbf{AP}_M$, and $\mathbf{AP}_L$ in Tab. 4. It can be seen that our method improves the AP performance on medium and large objects but shows a slight AP drop on small objects. This is presumably because we keep the middle and deep levels unchanged, while the shallow levels are equipped with our SlimHead. The inevitable loss of information in shallow levels leads to performance degradation of small objects. Nevertheless, our method largely improves the accuracy of medium/large objects. Further, we conduct a more comprehensive evaluation of

Table 4: Performance comparison of Slim-Head on objects with different scales.

| SlimHead | FPS | AP | $\mathbf{AP}_S$ | $\mathbf{AP}_M$ | $\mathbf{AP}_L$ |
|---|---|---|---|---|---|
| | 37.0 | 40.1 | 23.3 | 44.4 | 52.5 |
| ✓ | 38.0 | 41.4 | 22.7 | 45.4 | 55.9 |

accuracy at various object scales. We borrow the idea of zone evaluation (Zheng et al., 2023a). The evaluated object scale is set to $t = 0, 16, 32 \cdots, 320$. We evaluate the detector with the ground-truth objects and the detections if their box areas fall in the interval $[t^2, (t+16)^2]$. As shown in figure 6**Left**, our method can achieve comparable AP performance at the object scale of $32^2$. It degrades the AP at a very small object scale $t \leqslant 16$, i.e., tiny objects. In figure 6**Right**, one can see that our method shows superiority when the object scale $t > 32$. When $t < 32$, the two AP curves are tightly close together since the performance degradation is slight enough. This demonstrates our SlimHead has a positive impact on a wide range of object scales.

### 4.2 COMPARISON WITH STATE-OF-THE-ART HEAD NETWORKS

In this subsection, we compare our method with state-of-the-art head networks. Our SlimHead is built on the strong dense object detector TOOD (Feng et al., 2021). We set the level selector $K = 3$ and DCN Zhu et al. (2019) is applied to the first three layers of the head network. Except that the single level detector YOLOF adopts ResNet-101 backbone, all the others adopt ResNet-50 backbone. The training is conducted on the single-scale $1\times$ (12 epochs) training schedule, which is a classic training setting in the detection community. The results are reported in Tab. 5. One can see that our SlimHead lifts the AP score by $+0.9$ upon the strong baseline TOOD, even bringing an inference speedup. For the first time, we achieve $>43.0$ AP on COCO val2017 in dense object detection under the clean settings of ResNet-50 single-scale $1\times$ training, while maintaining an FPS of $>30$. It is

Table 5: Performance comparison with state-of-the-art head networks among dense object detectors on MS COCO val2017. FPS is measured on a single RTX 3090 GPU.

| Head Network | FPS | AP | $AP_{50}$ | $AP_{75}$ | $AP_S$ | $AP_M$ | $AP_L$ |
|---|---|---|---|---|---|---|---|
| RetinaNet (Lin et al., 2017b) | 35.9 | 36.5 | 55.4 | 39.1 | 20.4 | 40.3 | 48.1 |
| FCOS (Tian et al., 2019) | 36.9 | 38.7 | 57.4 | 41.8 | 22.9 | 42.5 | 50.1 |
| ATSS (Zhang et al., 2020) | 36.9 | 39.3 | 57.5 | 42.8 | 24.3 | 43.3 | 51.3 |
| Double-Head (Wu et al., 2020) | 1.5 | 40.1 | 59.4 | 43.5 | 22.9 | 43.6 | 52.9 |
| GFocal (Li et al., 2020) | **37.0** | 40.1 | 58.2 | 43.1 | 23.3 | 44.4 | 52.5 |
| PAA (Kim & Lee, 2020) | 16.0 | 40.4 | 58.4 | 43.9 | 22.9 | 44.3 | 54.0 |
| AutoAssign (Zhu et al., 2020) | 34.5 | 40.4 | 59.6 | 43.7 | 22.7 | 44.1 | 52.9 |
| YOLOF (Chen et al., 2021a) | 29.5 | 40.5 | 59.8 | 43.9 | 23.0 | 44.9 | 53.8 |
| OTA (Ge et al., 2021) | 36.9 | 40.7 | 58.4 | 44.3 | 23.2 | 45.0 | 53.6 |
| GFocalV2 (Li et al., 2021) | 36.6 | 41.1 | 58.8 | 44.9 | 23.5 | 44.9 | 53.3 |
| VFNet (Zhang et al., 2021) | 31.2 | 41.5 | 59.1 | 45.2 | 24.4 | 45.4 | 53.9 |
| DDQ-FCN (Zhang et al., 2023) | 36.0 | 41.5 | **60.9** | 45.9 | 25.1 | 44.6 | 53.1 |
| DDOD (Chen et al., 2021b) | 36.7 | 41.6 | 55.9 | 45.1 | 23.4 | 44.8 | 55.3 |
| DW + box refine (Li et al., 2022b) | 35.0 | 42.1 | 59.9 | 45.1 | 24.2 | 45.3 | 55.9 |
| TOOD (Feng et al., 2021) | 33.6 | 42.3 | 59.6 | 45.9 | 25.8 | 45.6 | 54.9 |
| DyHead (Dai et al., 2021a) | 24.3 | 42.6 | 60.1 | 46.4 | **26.1** | 46.8 | 56.0 |
| **SlimHead** (Ours) | 34.2 | **43.2** | 60.4 | **47.0** | 24.2 | **47.0** | **58.5** |

Table 6: Detection performance on Faster R-CNN and PSC. For Faster R-CNN (Ren et al., 2015), the results are reported on COCO val2017. For PSC (Yu & Da, 2023), the results are reported on DOTA-v1.0.

| Detector | SlimHead | AP | $AP_{50}$ | $AP_{75}$ | FPS |
|---|---|---|---|---|---|
| Faster R-CNN | | 37.4 | 58.1 | 40.4 | 37.7 |
| | ✓ | 37.8 | 58.7 | 41.3 | 38.7 |
| PSC | | 41.9 | 68.2 | 42.9 | 31.3 |
| | ✓ | 43.1 | 68.8 | 43.9 | 30.1 |

Table 7: Detection performance on CrowdHuman (Shao et al., 2018) and PASCAL VOC (Evering-ham et al., 2010). For CrowdHuman, we apply DCN (Zhu et al., 2019) at the first layer, while it is the first three layers for VOC.

| Datasets | SlimHead | AP | $AP_{50}$ | $AP_{75}$ | FPS |
|---|---|---|---|---|---|
| VOC | | 56.3 | 79.3 | 62.0 | 33.6 |
| | ✓ | 57.7 | 80.3 | 63.5 | 34.2 |
| CrowdHuman | | 44.0 | 78.8 | 43.3 | 33.6 |
| | ✓ | 44.6 | 79.1 | 44.1 | 34.2 |

worth noting that although our method cannot be directly extended to the query-based detectors, e.g., DETR series (Carion et al., 2020; Zhu et al.) due to the significant structure differences among object detectors, Tab. 5 shows us promising results that **dense object detectors can still perform better than query-based detectors at the algorithmic level.** To be specific, DDQ-FCN (Zhang et al., 2023) adopts the FCOS-like structure but with query-based learning. It follows the same one-to-one bipartite matching as the DETR-based detector does. The results of Tab. 5 show that if the detection network is aligned between the dense object detector and the query-based detector, our SlimHead (43.2 AP) can still outperform query-based detector, i.e., DDQ-FCN (41.5 AP).

## 4.3 SLIMHEAD FOR OTHER OBJECT DETECTORS

In this subsection, we implement our SlimHead on 2 more popular object detectors. The first one is the representative multi-stage dense-to-sparse object detector Faster R-CNN (Ren et al., 2015). The second one is the recently popular arbitrary-oriented object detector PSC (Yu & Da, 2023). We use ResNet-50 backbone and FPN neck. For Faster R-CNN, we apply SlimHead on RPN with the same settings of $K = 3$ and DCN. For PSC, we apply SlimHead at the P4 level and DCN in the first conv layer. For evaluating PSC, the train / val sets are DOTA-v1.0 (Xia et al., 2018) train / val sets, respectively. As shown in Tab. 6, our method consistently boosts the detection performance of the two types of object detectors while maintaining high efficiency, demonstrating the good generalization ability of our method. Notice that our method does not show a speedup on PSC. This is presumably because the DOTA dataset contains more small objects and thus we implement SlimHead only at the P4 level. Nevertheless, our method achieves +1.2 AP on PSC.

Table 8: Performance comparison of SlimHead on two instance segmentation methods. The results are reported on MS COCO val2017. FPS is measured on a single RTX 3090 GPU.

| Model | SlimHead | box | | | mask | | | FPS |
|-------|----------|-----|------|------|------|------|------|-----|
| | | **AP** | **AP**$_{50}$ | **AP**$_{75}$ | **AP** | **AP**$_{50}$ | **AP**$_{75}$ | |
| BoxInst | | 39.6 | 58.5 | 42.9 | 31.1 | 53.1 | 31.7 | 27.8 |
| | ✓ | 40.5 | 59.1 | 43.9 | 31.8 | 53.7 | 32.6 | 28.1 |
| CondInst | | 39.3 | 58.3 | 42.4 | 35.7 | 56.2 | 38.1 | 29.3 |
| | ✓ | 40.9 | 59.6 | 44.2 | 36.9 | 57.4 | 39.5 | 29.7 |

Table 9: GPU memory usage and detection performance of SlimHead on 3 TOOD models. The results are reported on MS COCO val2017. FPS is measured on a single RTX 3090 GPU. **TS:** training schedule.

| Model | TS | SlimHead | Memory (MB) | Reduction | **AP** | **AP**$_{50}$ | **AP**$_{75}$ | FPS |
|-------|----|----------|-------------|-----------|--------|---------------|---------------|-----|
| ResNet-18 | 1× | | 2,105 | | 38.0 | 54.6 | 40.7 | 47.1 |
| | | ✓ | 1,788 | ↓ 15.1% | 39.1 | 55.4 | 42.4 | 47.4 |
| ResNet-50 | 1× | | 3,967 | | 42.3 | 59.6 | 45.9 | 33.6 |
| | | ✓ | 3,653 | ↓ 8.0% | 43.2 | 60.4 | 47.0 | 34.2 |
| Swin-L | 2× | | 6,518 | | 50.1 | 68.8 | 54.6 | 6.6 |
| | | ✓ | 6,198 | ↓ 4.9% | 50.6 | 69.3 | 54.7 | 6.7 |

## 4.4 SLIMHEAD FOR OTHER DATASETS

Thus far, we have shown the effectiveness of our SlimHead on MS COCO and DOTA. We further check out the generalizability of our method by conducting experiments on two more detection datasets. The first one is PASCAL VOC (Everingham et al., 2010). We use the classic VOC 07+12 protocol. We train the detectors for 12 epochs and the learning rate decreases by a factor of 10 after 9 epochs. For VOC, we set the level selector $K = 4$ and use DCN in the first three layers of the head networks. The second is the pedestrian detection dataset CrowdHuman (Shao et al., 2018) under crowded scenarios. The train set is CrowdHuman train and the evaluation set is CrowdHuman val. We train the detectors for 30 epochs and the learning rate decreases by a factor of 10 after 24 and 28 epochs. We adopt TOOD (Feng et al., 2021) with ResNet-50 backbone. We empirically found that our method does not work at the shallowest level on CrowdHuamn, but works well at the P4 level. For CrowdHuman, we only use DCN at the first layer of the head networks since we do not apply SlimHead at the shallowest level, i.e., the P3 level. As shown in Tab. 7, our SlimHead improves the detection performance on the two datasets. In the meantime, we keep a high model inference speed.

## 4.5 SLIMHEAD FOR INSTANCE SEGMENTATION

We further incorporate our SlimHead into instance segmentation methods. We use BoxInst (Tian et al., 2021) and CondInst (Tian et al., 2020) with ResNet-50 backbone and FPN neck. We follow the official training setting, in which the 1× training schedule is adopted. We set the level selector $K = 4$ and use DCN in all layers of the head network. The results are reported by box AP and mask AP on COCO val2017. As shown in Tab. 8, our method can clearly improve the box AP and the mask AP of the two instance segmentation methods. This indicates that our method has good generalization ability, which is not only beneficial for object detection but also for instance segmentation tasks. More importantly, our method does not produce an inference speed drop, but a slight speed-up. This once again demonstrates that our method is highly generalizable and cost-free in practice.

## 4.6 SLIMHEAD FOR SAVING GPU MEMORY USAGE

As discussed in section 3.2, the GPU memory usage can also be reduced since the feature map resolution of shallow levels decreases. Here, we report the train-time GPU memory usage, as shown in Tab. 9. The mini-batch size of each GPU is 2 images. The level selector is set as $K = 3$, i.e., we only apply SlimHead at the P3 level. One can see that our SlimHead reduces GPU memory usage by a large margin. GPU memory usage is reduced by 15.1%, 8.0%, 4.9% on ResNet-18, ResNet-50, Swin-L (Liu et al., 2021), respectively. It is worth mentioning that the reduction in GPU memory usage becomes significant as the model becomes lightweight. This demonstrates an important advantage of our method, namely that it can save GPU memory usage, which we believe is particularly helpful for low-end edge devices. In Tab. 9, we also report the detection accuracy along

with FPS on the 3 models, i.e., ResNet-18, ResNet-50, Swin-L. It can be seen that our SlimHead achieves consistent AP improvement on the three backbones. Also, it is efficient and FPS has been increased.

## 5 CONCLUSION

In this paper, we revisit the popular multi-level learning framework in dense object detection. Shallow levels are time-consuming, so we aim for minimal modification to achieve a better speed-accuracy trade-off. A natural outcome is SlimHead, a very simple, efficient, and generalizable head network, which further unleashes the potential of multi-level learning for dense object detectors. Our design follows a two stage principle: Slim and Fat, which has 4 advantages: 1) reducing computational complexity; 2) flexible combination with improved conv ops for better accuracy, e.g., DCN; 3) saving GPU memory usage; and 4) highly generalizable to various detectors. Extensive experiments on MS COCO, CrowdHuman, DOTA, PASCAL VOC, generic / arbitrary-oriented object detectors, and instance segmentation have demonstrated the high practical value of our method.

**Limitations.** As discussed in section 4.1, our method may show an adverse effect on tiny objects. To address this, we can prevent applying SlimHead to the shallowest level. We also acknowledge the limitation of our work that we did not extend our SlimHead to more complex detectors like query-based detectors, e.g., DETR series (Carion et al., 2020; Zhu et al.; Dai et al., 2021b; Meng et al., 2021; Liu et al.; Li et al., 2022a; Zhang et al.; Jia et al., 2023; Zong et al., 2023). As this might require more specialized designs to accommodate their structures due to the significant differences among object detectors.

**Broader impact.** Since our method is not designed for a specific application, it does not directly involve societal issues.

## 6 REPRODUCIBILITY STATEMENT

Transparency and reliability are crucial to our research. In this statement, we summarise the measures taken to facilitate the replication of our work and provide references to the relevant sections in the main paper and appendix.

**Source code.** We intend to make our source code, model weights, datasets, and a detailed tutorial available to the public following the paper's acceptance. It will allow the following researchers to access and utilize our code to reproduce our experiments and results. The detailed installation and execution instructions will be listed in "README.md."

**Experimental setup.** We provide the basic implementation information of our SlimHead in the beginning of section 4. Besides, we provide the pseudo codes of our method, the forward function of the head network. Kindly refer to the Appendix A.1.

We provide the above resources and references to ensure the reproducibility of our work. It enables fellow researchers to verify our method, We also welcome any inquiries or requests for further clarification on our methods.

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

# A APPENDIX

## A.1 HOW TO IMPLEMENT SLIMHEAD

Alg. 1 provides the pseudo-code of SlimHead forward procedure.

Our SlimHead is quite simple to implement, it can be easily integrated into any dense object detection pipeline in which multi-level learning is adopted.

## A.2 MORE APPROACH TO ALLEVIATE THE COMPUTATIONS OF MULTI-LEVEL LEARNING

Following the discussion in section 3.2, we conduct two more designs of head networks to alleviate the computations. The key idea is to reduce the computation burden of shallow levels. The first one is the effect of the number of channels. As shown in figure 7**Left**, we reduce half of the channels at $P_3$, $P_4$ levels. The second is that we gradually reduce the number of conv layers from the deep level to the shallow level, as shown in figure 7**Right**. The results are reported in figure 8. It can be seen that both designs can lower the GFLOPs by a large margin. However, they increase the model parameters of the head networks since the weights cannot be shared between different levels. One can see that our SlimHead achieves the best accuracy of 39.4 AP and the fastest inference speed of 46.2 FPS among the three designs. From the perspective of implementation, our SlimHead is also the easiest approach with minimal modifications to the original head networks. Therefore, in our main paper, we propose SlimHead as our final solution to the problem.

**Algorithm 1** SlimHead

```python
def forward_slimhead(x, K):
    """
    Args:
        x (tuple[Tensor]): Features from the upstream network, each is a 4D-tensor.
        K (integer): the level selector, K = {0, 1, 2, 3, 4}.
    Returns:
        tuple: Usually a tuple of classification scores and bbox prediction

        - cls_scores (list[Tensor]): Classification scores for all scale levels, each is a
            4D-tensor, and the channel number is num_classes.
        - bbox_preds (list[Tensor]): Box logits for all scale levels, each is a 4D-tensor,
    """

    # Output class score list and bbox list
    cls_scores_list = []
    bbox_preds_list = []

    for idx, feat in enumerate(x):

        # Slim Stage:
        if idx < K: # Select shallow levels. idx = 0, 1, 2, 3, 4 corresponds to P3, P4, P5
            , P6, P7 levels.
            shape = feat.shape[2:]
            feat = F.interpolate(feat, scale_factor=0.5, mode='nearest')

        cls_feat = feat
        reg_feat = feat

        # Forward head network functions: 2 branches, 1 for classification, 1 for
            localization.
        for cls_conv in self.cls_convs:
            cls_feat = cls_conv(cls_feat) # When applying DCN, some of cls_conv and
                reg_conv will be DCN.
        for reg_conv in self.reg_convs:
            reg_feat = reg_conv(reg_feat)

        # Fat Stage:
        if idx < K: # Select shallow levels
            cls_feat = F.interpolate(cls_feat, size=shape, mode='nearest')
            reg_feat = F.interpolate(reg_feat, size=shape, mode='nearest')

        # Projection layers
        cls_score = self.projection_layer_cls(cls_feat)
        bbox_pred = self.projection_layer_reg(reg_feat)

        cls_scores_list.append(cls_score)
        bbox_preds_list.append(bbox_pred)

    return cls_scores_list, bbox_preds_list
```

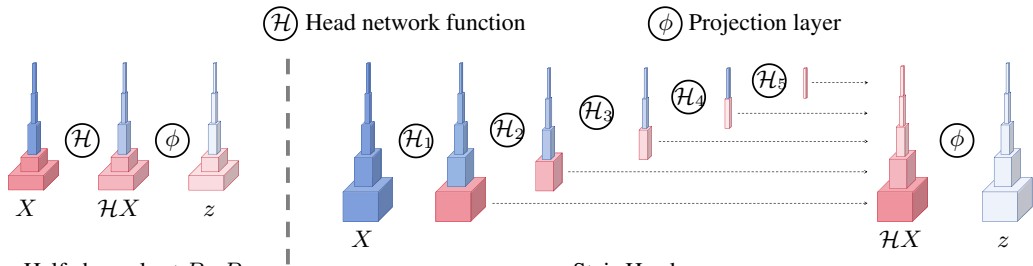

Figure 7: **Left**. Reduce half of channels at $P_3, P_4$ in Plain Head. **Right**. Stair-Head: Set 1, 2, 3, 4, and 5 conv layers for $P_3$ to $P_7$, respectively. For Stair-Head, the head network function $\mathcal{H}$ consists of 5 conv ops $\mathcal{H}_1, \mathcal{H}_2, \cdots, \mathcal{H}_5$. Different colors indicate that the conv layers do not share weight. $X$: Features from the upstream network, e.g., backbone & FPN. $z$: Output logits for both class scores and bounding box coordinates.

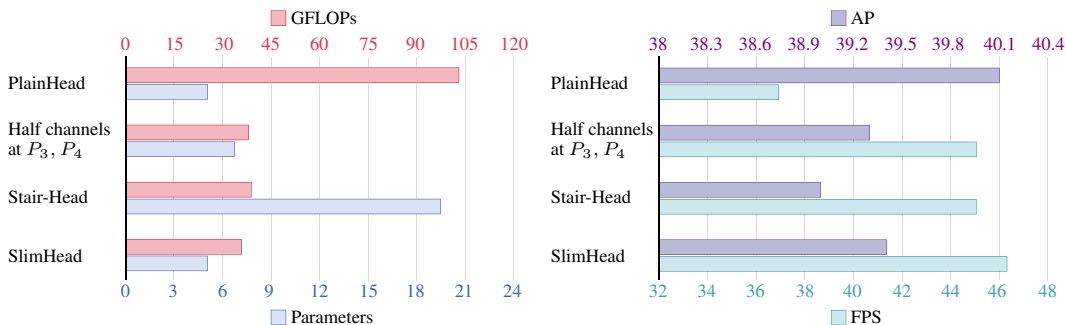

Figure 8: GFLOPs, model parameters (M), accuracy (AP), and speed (FPS) comparison between various head network designs. Our SlimHead notablely reduces the computational complexity of the head networks while keeping consistent model parameters. Meanwhile, we achieve the best accuracy of 39.4 AP and 46.4 FPS among these designs.

