# OpenReview forum: "Efficient Multi-Level Learning for Dense Object Detection"
_ICLR.cc/2025/Conference — ICLR 2025 Conference Withdrawn Submission_

### Official Review · Reviewer_DU2A · 2024-10-23

**Soundness:** 2
**Presentation:** 2
**Contribution:** 2
**Rating:** 3
**Confidence:** 5

**Summary:**

In this paper, the authors aim for minimal modifications to exchange a better speed-accuracy trade-off. The outcome is SlimHead, a very simple, efficient, and generalizable head network, which further unleashes the potential of multi-level learning for dense object detectors.

**Strengths:**

* The method is easy to understand.

**Weaknesses:**

* The novelty is limited. It seems the method only applied an interpolator&inversed interpolator into layers to change the size of feature map. In my opinion, it's just an engineering trick.
* The experiment are seriously out of date. The method achieves 43.2 on MS COCO2017. But according to the [leading board](https://paperswithcode.com/sota/object-detection-on-coco-minival), the detectors now can achieve over 65.9 now. Even taking into account the small backbone network chosen for the experiment, the results are only comparable to the level of research conducted a few years ago. I don't think a small boost is enough in such a weak experimental setup: for example, according to Table 6, the proposed method only improves 0.4 on the Faster R-CNN.
* The comparison is problematic. It seems the performance of other works is much lower than the official report. For example, in Table 5, GFocalV2 achieves 41.1 on the COCO 2017 val set, but according to the [official paper](https://github.com/implus/GFocalV2), GFocalV2 with the R50 backbone performs 44.3 test set. Although there are differences between the two datasets, there is generally not such a large performance loss.

**Questions:**

The authors may:

1.  Applied their method on the latest works and dig further into the innovations of the methodology.
2. Explain why the performance of other works in the Table 5 is lower than the official results.

---

> ### Author Response · Authors · 2024-11-12
> **Rebuttal to reviewer**
>
> We appreciate your time spend in reviewing our paper. The following is our response to your comments. And we are open to communicate with the reviewer. Please feel free to leave your comment so that more sparks of thought would be generated.
>
> **Answer to weakness 1**: We acknowledge that this paper is not for introducing new technical component for object detection. As we state in Abstract, '**'we aim for minimal modifications to exchange a better speed-accuracy trade-off**.'' What we did in this paper is to explore new insights on the intrinsic property of head networks and how can we make full use of them for efficient object detection. In our design ethos, ''**simplicity**'' is the primary consideration and we did not expect building a sophisticated method. Maybe it looks like an engineering trick. It is important to note that even though general object detectors are highly developed today, **the nature of the detection head is still not well explored**. Based on the results from our paper, all we need is to keep high resolution for the last conv layer of the head network. It could be an interesting finding that it may overturn all the designs before the final layer, as long as we keep the size of the prediction map for the last layer. That's what we believe is inspiring and promising.
>
> **Answer to weakness 2**: As an academic paper, it is a common sense that pursuing state-of-the-art accuracy on leaderboard is not necessary for getting an acceptance. Also, to our best knowledge, under the classic settings of ResNet50-FPN $1\times$ single-scale training schedule, the proposed SlimHead achieves state-of-the-art performance among all existing head networks in dense object detection (under fair comparison).
>
> > Why ''the results are only comparable to the level of research conducted a few years ago''?
>
> Because this is the current research state in head networks. As a side evidence, in knowledge distillation object detection, the current state-of-the-art dense object detector achieves 43.9 AP [r1]. This is achieved by distilling the knowledge from a well-pretrained teacher model (ResNet101-FPN multi-scale $2\times$ training schedule). It can be clearly seen that our result of 43.2 AP is quite high that is even close to the result of knowledge distillation.
>
> For the improvement on Faster R-CNN, while a 0.4 AP gains is small, it is noted that AP$_{75}$ is improved by +0.9, and even getting a 1.0 FPS gains. Nevertheless, we achieve more gains (>0.4) under many scenarios, e.g., rotated object detector (PSC), two detection datasets (VOC and CrowdHuman), and two instance segmentation methods (BoxInst, CondInst), etc.
>
> **Answer to weakness 3**: The training schedule is different. The results 41.1 of GFocalV2 is under single-scale $1\times$ (12 epochs) training schedule, while 44.3 (test-dev) is under $2\times$ (24 epochs) training schedule. Kindly check the results of GFocalV2 paper [r2].
>
> **Answer to question 1**: Thanks the reviewer. We will continue to delve deep into the head network for object detection.
>
> **Answer to question 2**: See Answer to weakness 3.
>
> [r1] Wang J, Chen Y, Zheng Z, et al. CrossKD: Cross-head knowledge distillation for object detection. CVPR 2024.
>
> [r2] Li X, Wang W, Hu X, et al. Generalized focal loss v2: Learning reliable localization quality estimation for dense object detection. CVPR. 2021.

---

### Official Review · Reviewer_iL3L · 2024-10-30

**Soundness:** 3
**Presentation:** 3
**Contribution:** 2
**Rating:** 5
**Confidence:** 3

**Summary:**

This paper introduces SlimHead, a head network that optimizes speed and accuracy with minimal modifications. SlimHead processes feature pyramids in two stages—first "slimming" and then "fatting" the features—allowing for complex operations without sacrificing inference efficiency. This adaptable design improves accuracy across tasks like object and pedestrian detection and instance segmentation, as shown through extensive experiments on datasets such as PASCAL VOC, MS COCO, DOTA, and CrowdHuman.

**Strengths:**

- The paper is clearly written.
- The ablation study sufficiently demonstrates the effectiveness of each component.
- Several instance-level perception tasks are performed.

**Weaknesses:**

- The comparison methods in Tab. 5 for object detection on MS COCO val2017 are outdated.
- More real-time methods【1】【2】should be included for comparison.
- Given that the main technical contribution is to achieve a better speed-accuracy trade-off for object detection, the current results are insufficient to demonstrate this.

[1] Zhao, Yian, et al. "Detrs beat yolos on real-time object detection." Proceedings of the IEEE/CVF Conference on Computer Vision and Pattern Recognition. 2024.

[2] Wang, Ao, et al. "Yolov10: Real-time end-to-end object detection." arXiv preprint arXiv:2405.14458 (2024).

**Questions:**

Please see the above weaknesses.

---

> ### Author Response · Authors · 2024-11-12
> **Rebuttal to reviewer**
>
> We appreciate your time spend in reviewing our paper. The following is our response to your comments. And we are open to communicate with the reviewer. Please feel free to leave your comment so that more sparks of thought would be generated.
>
>
> **Answer to weakness 1**: To our best knowledge, under the classic settings of ResNet50-FPN $1\times$ single-scale training schedule, the proposed SlimHead achieves state-of-the-art performance among all existing head networks in dense object detection (under fair comparison).
>
> > Why ''the comparison methods in Tab. 5 for object detection on MS COCO val2017 looks outdated''?
>
> Because this is the current research state in head networks. As a side evidence, in knowledge distillation object detection, the current state-of-the-art dense object detector achieves 43.9 AP [r1]. This is achieved by distilling the knowledge from a well-pretrained teacher model (ResNet101-FPN multi-scale $2\times$ training schedule). It can be clearly seen that our result of 43.2 AP is quite high that is even close to the result of knowledge distillation.
>
> **Answer to weakness 2**: If we are correct, RT-DETR is DETR-based detectors, not dense object detector. As we state in the Limitation section, ''We also acknowledge the limitation of our work that we did not extend our SlimHead to more complex detectors like
> query-based detectors, e.g., DETR series. As this might require more specialized designs to accommodate their structures due to the significant differences among object detectors.'' For YOLOv10, this paper is recently accepted by NeurIPS 2024 which is just a few days ago before the submission deadline of ICLR 2025. We sincerely argue that there is no need to include it.
>
> **Answer to weakness 3**: It can be clearly seen in Tab.5,6,7,8,9 that our method improves the detection accuracy and achieve an inference speed-up. We have demonstrated the effectiveness of the our method on 4 detection datasets (MS COCO, CrowdHuman, DOTA, PASCAL VOC), 3 high-level vision tasks (generic / arbitrary-oriented object detection and instance segmentation). As you recognized in Strength part, **"The ablation study sufficiently demonstrates the effectiveness of each component" and "Several instance-level perception tasks are performed,** it can be concluded that the current results have sufficiently demonstrated that our method can achieve a better speed-accuracy trade-off for dense object detection.
>
> [r1] Wang J, Chen Y, Zheng Z, et al. CrossKD: Cross-head knowledge distillation for object detection. CVPR 2024.

---

### Official Review · Reviewer_U8dh · 2024-11-03

**Soundness:** 2
**Presentation:** 3
**Contribution:** 2
**Rating:** 5
**Confidence:** 5

**Summary:**

The paper proposes a generic head module named SlimHead to reduce the heavy computational cost of dense object detectors at shallow feature levels by reducing the spatial size by a factor of r. Experiments are conducted on COCO, DOTA, and PascalVOC datasets and on different detectors such as TOOD, Faster-RCNN, and Phase-shifting coder.

**Strengths:**

The paper proposes an efficient head module to improve the speed-accuracy trade-off for dense object detection architectures. Experiments are conducted across different datasets.

**Weaknesses:**

The novelty of the method is limited as it is known that operating in the latent space reduces the cost of computations and the method still requires retraining existing architectures to improve its efficiency.

**Questions:**

Will the method work if you finetune only the head layers from pretrained models using a subset of data? It can then apply to a broader class of models.

Inaccuracy in Table 5: TOOD paper has different values than reported in Table 5. 42.5 (original) vs 42.3 (here). Can the authors explain the discrepancy?

In general, the speed only improves marginally (< 1 FPS) for all the models shown. How does it compare with efficient backbones such as efficientnetv2 or efficientvit? Since, the paper deals with efficiency it should compare against these baselines.
Further, how does it compare with quantization methods that do not require retraining or pruning methods [1,2]?

The performance improvement is mainly observed across medium and large-scale objects. Why do they improve when you only modify the shallow features (for small objects)? Is there a trade-off between the different feature scales?

The claim of a generic module is not supported by comprehensive experiments: Experiments are shown with only the ResNet-50 backbone. Further, Table 9 shows that the performance improvement with larger models like Swin-L is small (even with 2x schedule) across both speed and accuracy. So, additional experiments with ResNet101 backbone and other architectures like RetinaNet are needed to claim it is a generic module.

The configuration of Slimhead is not uniform and is engineered for different tasks and datasets (Sections 4.3, 4.4, 4.5) as DCN is used for the first two layers for object detection and in all layers, for instance segmentation. How to choose the configurations, does it require extensive hyperparameter tuning? Can it be automated with Neural Architecture Search?

For the alternatives, have the authors considered reducing both spatial and channel dimensions? Have you considered residual connections in SlimHead?

Minor:
The writing of the paper can be improved further – e.g., sub-figures are denoted as figure3Right with Right in caps. Figure 3 can be improved with marked arrows.
Typo in L214 – “dimention”

[1] Molchanov et. al. Pruning convolutional neural networks for resource efficient inference, ICLR 17
[2] He et. al., Channel pruning for accelerating very deep neural networks. ICCV 17

---

### Official Review · Reviewer_7k7x · 2024-11-04

**Soundness:** 3
**Presentation:** 3
**Contribution:** 2
**Rating:** 3
**Confidence:** 5

**Summary:**

The paper proposes a 'slim' and 'fat' strategy to reduce the computational burden of high-resolution features in multi-level object detection. The 'slim' step reduces the feature resolution by interpolation, and the 'fat' step recovers the original resolution. By incorporating DCN, the resulting model achieves slightly better accuracy and inference speed than the baseline model. The paper also presents experimental results on pedestrian detection and instance segmentation.

**Strengths:**

1. The writing and presentation are good, especially the figures in the paper.
2. Extensive experiments were conducted to analyze the effectiveness of the proposed method.

**Weaknesses:**

1. Limited novelty. The proposed method is simple, and the performance improvements are minimal. It appears to be more of an engineering technique than a sufficient method for publication at ICLR.

2. Multiple alternative approaches can achieve similar improvements. An intuitive and simpler method would be to reduce the channels of low-resolution features. The paper already presents this result in Figure 8. The model with half the channels at P3 and P4 achieved comparable results to the proposed method (AP 39.3 vs. 39.4, FPS 39.9 vs. 40.2). This approach adds 2M parameters, but this is negligible compared to the entire model. Why not adopt this approach to achieve similar results?

3. Improving the inference speed of the detection head has less impact when using a larger backbone. The results in Table 9 illustrate this (when using Swin-L as the backbone, FPS 6.6 vs. 6.7). With the rapid development of LLMs, such an improvement becomes less significant.

4. Another way to enhance detection accuracy with minimal computational cost is to apply DCN only on high-level features, as DCN mainly benefits large objects (Table 4). This may be a more cost-effective solution than the proposed method, offering similar detection accuracy with faster inference speed.

In summary, there are many engineering techniques that can achieve similar results to the methods in the paper. The proposed method offers no clear advantages over these alternatives.

**Questions:**

See the weaknesses.

---

> ### Author Response · Authors · 2024-11-12
> **Rebuttal to reviewer**
>
> We appreciate your time spend in reviewing our paper. The following is our response to your comments. And we are open to communicate with the reviewer. Please feel free to leave your comment so that more sparks of thought would be generated.
>
> **Answer to weakness 1**: We acknowledge that this paper is not for introducing new technical component for object detection. As we state in Abstract, '**'we aim for minimal modifications to exchange a better speed-accuracy trade-off**.'' What we did in this paper is to explore new insights on the intrinsic property of head networks and how can we make full use of them for efficient object detection. In our design ethos, ''**simplicity**'' is the primary consideration and we did not expect building a sophisticated method. Maybe it looks like an engineering trick. It is important to note that even though general object detectors are highly developed today, **the nature of the detection head is still not well explored**. Based on the results from our paper, all we need is to keep high resolution for the last conv layer of the head network. It could be an interesting finding that it may overturn all the designs before the final layer, as long as we keep the size of the prediction map for the last layer. That's what we believe is inspiring and promising.
>
> **Answer to weakness 2 and 4**: In the paper, we try many solutions for exchange a better speed-accuracy trade-off. Among all the solutions, SlimHead is just the most efficient. But it should be noted that all the alternative approaches share a common point, i.e., **the size of the prediction maps keep consistent as normal.** This is what we explore in the paper that the key to maintain comparable accuracy but not a significant performance drop is to maintain the size of predictions for the last conv layer. Only if we keep this, these alternative approaches can work. Then, Slimhead is the most effective one. We argue that this will not weaken the key contribution of the paper.
>
> **Answer to weakness 3**: Yes, it is true that with the backbone network size increase, our method produces less positive impact. To be honest, this phenomenon is quite common in the field of object detection. But still, we achieve +0.5 AP for Swin-L backbone, even with 4.9% GPU memory usage reduction.

---

### Note · Authors · 2024-11-19

**Comment:**

We would like to thank reviewers for providing constructive feedback. We are encouraged that reviewers think our paper is well-written with clear presentation or easy to understand. We also appreciate that the reviewers found the method is simple and effective, although there are still many scenarios that we may not be able to cover. It seems that the main concern about the limited novelty is coming from the simplicity. We have to admit that the proposed method in its current form is the final version after extensive trial and error. And it is indeed simple than we can ever imagine. Thanks to its simplicity and flexibility, our SlimHead offers more possibility of improvement for dense object detection such as better accuracy, faster speed, and even GPU memory usage reduction. The experiments on 4 detection datasets, 3 high-level vision tasks verify the effectiveness of the method. During the discussion time being, we acknowledge that some of answers may not satisfy the reviewers and there are still several questions that we may not able to provide the answers:

1) Applying SlimHead upon EfficientNetv2 or EfficientViT backbones.

2) Comparison with some quantization methods.

After consideration, we decide to withdraw the submission from ICLR 2025. Thanks again to each reviewer for their efforts.

**Withdrawal Confirmation:**

I have read and agree with the venue's withdrawal policy on behalf of myself and my co-authors.